# An Explainable AI Exploration of the Machine Learning Classification of Neoplastic Intracerebral Hemorrhage from Non-Contrast CT

**DOI:** 10.3390/cancers17152502

**Published:** 2025-07-29

**Authors:** Sophia Schulze-Weddige, Georg Lukas Baumgärtner, Tobias Orth, Anna Tietze, Michael Scheel, David Wasilewski, Mike P. Wattjes, Uta Hanning, Helge Kniep, Tobias Penzkofer, Jawed Nawabi

**Affiliations:** 1Department of Radiology, Campus Virchow, Charité—Universitätsmedizin Berlin, Humboldt-Universität zu Berlin, Freie Universität Berlin, Berlin Institute of Health, 13353 Berlin, Germany; georg.baumgaertner@charite.de (G.L.B.); tobias.orth@charite.de (T.O.); tobias.penzkofer@charite.de (T.P.); 2Department of Neuroradiology, Campus Mitte, Charité—Universitätsmedizin Berlin, Humboldt-Universität zu Berlin, Freie Universität Berlin, Berlin Institute of Health, 10117 Berlin, Germany; anna.tietze@charite.de (A.T.); michael.scheel@charite.de (M.S.); mike.wattjes@charite.de (M.P.W.); jawed.nawabi@charite.de (J.N.); 3Department of Neurosurgery, Campus Mitte, Charité—Universitätsmedizin Berlin, Humboldt-Universität zu Berlin, Freie Universität Berlin, Berlin Institute of Health, 10117 Berlin, Germany; david.wasilewski@med.uni-duesseldorf.de; 4Department of Neuroradiology, University Medical Center Hamburg-Eppendorf, 20246 Hamburg, Germany; u.hanning@uke.de (U.H.); h.kniep@uke.de (H.K.); 5Berlin Institute of Health (BIH), BIH Biomedical Innovation Academy, 10117 Berlin, Germany

**Keywords:** intracerebral hemorrhage, brain edema, computed tomography scanner, X-ray, artificial intelligence, machine learning

## Abstract

This study investigates which imaging features a deep-learning model uses to distinguish between neoplastic and non-neoplastic brain hemorrhages. Explainable artificial intelligence techniques show that the model relies primarily on features in the hemorrhage, but also considers features in the surrounding edema.

## 1. Introduction

Intracerebral hemorrhage (ICH) associated with primary and metastatic brain tumors presents a significant challenge in neuro-oncology due to the substantial risk of complications [1]. A major contributor to this challenge is the diagnostic complexity, particularly during the early stages of presentation [2]. Patients with tumor-related ICH often exhibit symptoms resembling, among others, those of spontaneous hypertensive hemorrhages, which can frequently serve as the initial clinical manifestation preceding tumor-specific symptoms [2,3,4,5]. This similarity can make differentiation based on clinical and imaging findings difficult, potentially delaying the initiation of etiology-specific work-up protocols. These delays may not only impact prognosis and therapeutic outcomes but also result in unnecessary diagnostic procedures, raising concerns about both clinical and economic efficiency [6,7,8,9,10]. Accurate and early detection of neoplastic ICH is, therefore, essential.

In most cases, computed tomography (CT) imaging remains the gold standard, particularly as these patients often present in acute clinical settings. Recent studies, have proposed various quantitative approaches to leverage perihematomal edema (PHE) characteristics surrounding the hemorrhagic lesion to differentiate neoplastic from non-neoplastic hemorrhages [11,12,13,14]. Notably, in our previous work, we introduced an end-to-end deep learning approach with significant potential for clinical translation [15]. This method combines an automated segmentation model to delineate lesions of interest with a classification model to distinguish neoplastic from non-neoplastic ICH, thereby eliminating the need for manual segmentation.

In the current study, we do not modify or retrain this classification model. Instead, we focus on analyzing how the previously trained model arrives at its predictions. Despite its strong performance, the model—like many deep neural networks—functions as a “black box,” making it difficult to interpret and trust in clinical settings. To address this challenge, we apply post hoc explainable artificial intelligence (XAI) techniques to enhance the transparency and interpretability of the model’s decision-making process [16].

Our hypothesis is that feature attribution methods, which provide pixel-wise significance scores to highlight critical regions, will offer valuable insights into the model’s inner workings and reaffirm the predictive importance of the PHE region in distinguishing between neoplastic and non-neoplastic ICH. To test this hypothesis, we compared the average importance attributed to ICH and PHE regions in the classification process.

## 2. Methods

### 2.1. Study Population

Our study consisted of two retrospectively assembled patient cohorts. The first cohort was gathered from Charité University Hospital Berlin, Germany, from January 2016 to May 2020. The second cohort included patients from a further academic hospital, the University Medical Center Hamburg-Eppendorf, Germany, from January 2010 to December 2017. For the purpose of this study, these two cohorts were pooled together to create a larger and more diverse dataset for the evaluation of the explainability methods. The inclusion criteria were consistent across both cohorts, requiring patients to have an ICH diagnosis on CT imaging, followed by MRI imaging. Cases were categorized into non-neoplastic and neoplastic ICH based on the H-ATOMIC classification [17]. Patient characteristics can be found in Table 1. Illustrative cases for neoplastic and non-neoplastic cases can be found in Figure 1.

### 2.2. Image Analysis and Preprocessing

Non-contrast CT images were retrieved from the local picture archiving and communication system (PACS) servers, anonymized in line with local protocols, and converted to Neuroimaging Informatics Technology Initiative (NifTI) format. Semi-manual planimetric measurements quantified the extent of ICH and PHE. For both cohorts, this analysis was performed by a trained research student or radiology resident with three years of experience in ICH imaging. Additionally, all cases were reviewed by a radiology fellow with eight years of experience in ICH imaging. In case of discrepancies, a consensus reading was performed, as detailed in our previous studies [13,14]. The clinical data and radiological reports were blinded during image analysis. Segmentations were used to mask the CT images, and the images were cropped from the original size of (512, 512, 31) to (200, 200, 20). This simplification substantially reduced the classification task’s complexity, allowing the model to concentrate on relevant image regions.

### 2.3. Classification Pipeline

In this study, we used XAI to compare the importance of different imaging features in the machine learning-based classification of neoplastic and non-neoplastic ICH, which is based on a previously trained and externally validated classification model [15]. In the original work, a residual neural network (ResNet) model was trained with preprocessed images for the classification of neoplastic and non-neoplastic ICH. The preprocessing entailed a segmentation of the ICH and PHE regions. This segmentation was automatized with an nnU-Net segmentation model. The two models were integrated in an end-to-end pipeline in which the automatically generated segmentations were used to preprocess the images for the classification task. A graphical representation of the workflow is illustrated in Figure 2. The classification model yielded an area under the curve (AUC) of 83% with an accuracy of 80%, sensitivity of 72%, and specificity of 89% on the full study population. Details about the model training can be found in the Appendix A or in the original publication [15].

### 2.4. Explanation Methods

Explanations were generated for 349 cases (144 neoplastic, 205 non-neoplastic). Various established explanation methods have been applied, namely Saliency [18], InputXGradient [19], SmoothGrad [20], Gradient Shap [21], GradCam [22], Guided GradCam [22], and GradCam++ [23]. All employed methods are primary attribution methods determining the importance of individual input features on the output. Each method returns an attribution map the same size as the input image. Each pixel value in the attribution map corresponds to the importance of that pixel for the prediction. The explanations are local, meaning they do not explain the model behavior in general but indicate what was important for the prediction of the specific input instance.

The methods visually differ in the granularity and smoothness of the highlighted regions. Saliency and InputXGradient are first-order gradient-based attribution methods that calculate the gradients of the output with respect to the input. They typically result in sharp, sometimes noisy attribution maps. SmoothGrad and Gradient Shap offer more stable and smooth attribution maps that are less deceptive to noisy inputs. SmoothGrad achieves this by adding noise to the input image and averaging over the resulting attribution maps. Gradient Shap integrates over multiple baselines to estimate feature importance, combining ideas from Shapley values with gradient-based methods. The GradCam versions (GradCAM, Guided GradCAM, GradCAM++) focus more on high-level features and broader areas of importance rather than individual pixels. GradCam++ considers positive and negative gradients separately, which helps in preserving more precise localization information and generating sharper heatmaps.

A comprehensive description of the methods and their differences has been provided in the Appendix A. Figure 3 shows an exemplary case for each method. The mean importance of the ICH and PHE region was calculated by averaging the importance of all pixels belonging to that region.

### 2.5. Faithfulness Metric

The attribution methods were quantitively evaluated using the faithfulness metric, which measures the relevance of selected features for the model’s prediction [24]. The metric is obtained by calculating the correlation coefficient between the attribution value of each pixel and the change in prediction probability when the pixel is replaced by a value from a baseline image. This baseline represents the absence of information and is usually challenging to determine. Common baseline values include zero, the average value of the image, or a random value sampled from the pixel distribution. In this study, a baseline of zeros was chosen consistent with the background masking applied during preprocessing.

Iterating through all pixels of the image, a prediction is made on the image with this pixel value set to zero. The model’s prediction probability for the target class is observed. In case the pixel was important for the prediction, a drop in prediction probability is expected. A high correlation between the changes in prediction probability and the attribution values indicates that the attribution map reflects the model’s decision-making process well.

The metric produces a correlation coefficient with possible values ranging from −1 to 1. Positive values indicate a positive linear relation between the variables. In this case, it means that with increasing attribution values, the prediction probability also increases, which is interpreted as indicating a good explanation. Negative values indicate that there is a negative linear relation between the variables, meaning if the attribution value increases, the prediction probability decreases. A value of 0 means there is no linear relation between the two variables. Values of ±0.1, ±0.3, and ±0.5 typically represent small, medium, and high correlations, respectively, which facilitates the evaluation of the explanatory quality. However, there is no established threshold for a “good” explanation. In this study, we used the faithfulness metric to benchmark and discern the most effective explanation method for our model, without the need for a definitive threshold. The scores of each method are provided in Table 2 and may serve as a reference for future research or comparisons.

### 2.6. Statistical Analysis

Data was tested for normality with the Shapiro–Wilk test. If the assumption of normality was met, variables were compared with a two-sided *t*-test, if not with the Mann–Whitney-U test. All two-sided hypothesis tests were considered statistically significant with a level of *p* < 0.05. The average importance of the ICH region was compared with the average importance of the PHE region. Further, the average ICH and PHE importances were compared between the neoplastic and non-neoplastic cases. Additionally, this comparison was performed for small and large lesions separately, with the median lesion volume separating the two subgroups.

## 3. Results

We generated and compared explanations for 349 cases (144 neoplastic, 205 non-neoplastic). The median ICH volume was 6.92 mL (IQR 2.21–19.91). There was no significant difference in ICH volume between the two classes (see Table 1). The distribution of ICH volumes for both classes is visualized in a histogram plot in Figure 4.

The explanation method with the highest faithfulness scores was GradCam++, with an average of 0.49 and standard deviation of 0.15. Hence, the attribution methods from GradCam++ were used to compare the average importance of ICH and PHE in neoplastic and non-neoplastic cases. The scores for all explanation methods can be found in Table 2. The overall mean importance was 0.639 for ICH and 0.435 for PHE, compared to 0.014 for the background (BG). Separated by class, the mean importance for ICH was 0.663 in non-neoplastic cases and 0.615 in neoplastic cases, whereas for PHE, they were 0.439 and 0.430, respectively, as detailed in Table 3. The distribution of importance scores is illustrated in a violin plot in Figure 5.

The statistical analysis of the full study population, as well as for the small and large lesions separately, revealed significant differences: (1) between the mean importance of ICH and PHE (all *p* < 0.001), (2) between the mean importance of ICH in the neoplastic and non-neoplastic group (all *p* < 0.01), and (3) between the mean BG importance in the neoplastic and non-neoplastic group (all *p* < 0.001), as detailed in Table 3. No significant difference was found in the mean importance of PHE between the two groups for the full study population (*p* = 0.54). However, when separated by lesion volume, there was a significant difference. In large lesions, the average PHE importance was higher in neoplastic cases compared to non-neoplastic cases (*p* = 0.001). The opposite was true for the small lesions, in which the average PHE importance was lower for the neoplastic cases (*p* = 0.02).

## 4. Discussion

The predictions of a convolutional neural network (CNN) for the binary classification of neoplastic and non-neoplastic ICH have been explained with the GradCam++ attribution method to gain insights into the inner working of the model and to confirm the importance of PHE for the classification task. An early and accurate differentiation of ICH types is important, as it enables timely and appropriate treatment decisions, which are essential for improving survival rates and reducing long-term disability. Gaining insights into the model by analyzing the average importance of ICH and PHE regions is important because it helps validate the model’s decision-making process and ensures that it aligns with clinical knowledge.

By understanding how the model uses these regions to differentiate between neoplastic and non-neoplastic ICH, we can confirm that the model is focusing on relevant anatomical features, increasing our confidence in its predictions. This transparency enhances the trustworthiness of the model in clinical practice, as it demonstrates that the AI model is making decisions based on meaningful and clinically significant information.

The generated explanations showed that both PHE and ICH were important for the differentiation of neoplastic and non-neoplastic ICH on admission CT. Yet, the ICH region consistently showed a higher average importance than the PHE region in both classes, irrespective of lesion volume. This suggests that the model’s differentiation between the two classes relied less on variations in PHE volume than initially hypothesized. We acknowledge that this finding appears to contrast with previous work, including our own earlier studies that demonstrated the diagnostic value of PHE volume in distinguishing neoplastic from non-neoplastic ICH [12,14]. This highlights important differences between traditional feature-based approaches and deep learning models. Specifically, while prior studies analyzed manually extracted imaging metrics (e.g., absolute and relative PHE volume), the deep learning model used here may prioritize more subtle textural and density cues embedded within the ICH region itself. One possible mechanistic explanation lies in the lower density of neoplastic ICH on CT images compared to non-neoplastic ones. The reasons for the lower density are likely the presence of intermixed tumor tissue and the tumor’s slower hemorrhage compared to abrupt ruptures in hypertensive associated bleedings [11]. This highlights density as a key predictive factor in classifying ICH etiology, which is supported by the significant differences in importance between neoplastic and non-neoplastic cases for ICH (*p* < 0.001). Our findings did not corroborate the anticipated higher discriminatory power of PHE in the automated classification. Although PHE contributed to the classification process, its importance did not outperform that of ICH.

Interestingly, a significant difference in PHE importance emerged when separating the cohort into lesions smaller and larger than the median ICH volume of 6.92 mL. Our results demonstrated that PHE importance was higher in neoplastic cases only for large lesions, whereas for smaller lesions, PHE importance was lower compared to non-neoplastic cases. This observation underscores the pathophysiological and temporal differences in edema formation between tumor-related and spontaneous hemorrhagic lesions, offering insights into the discriminative capabilities of our XAI approach.

Larger neoplastic ICHs are likely associated with a longer duration of tumor growth, which could lead to a larger preexisting vasogenic PHE. Vasogenic edema is driven by the disruption of the blood–brain barrier, permitting the accumulation of protein-rich fluid in the extracellular space [25]. This phenomenon is particularly prominent in larger tumors, which are associated with greater angiogenesis and vascular permeability [26,27]. Additionally, larger tumors exert a more substantial mass effect, further exacerbating blood–brain barrier disruption and enhancing edema formation [28]. Thus, the pronounced importance of PHE in larger neoplastic lesions detected by our model likely reflects these underlying tumor-related mechanisms.

In contrast, smaller neoplastic hemorrhages may not have had sufficient time or tumor activity to generate significant vasogenic edema. Consequently, the PHE observed in such cases might predominantly result from the acute hemorrhagic insult itself. Early PHE formation within the first four hours post-hemorrhage is primarily osmotic in nature, driven by clot retraction and the release of plasma proteins, rather than tumor-specific mechanisms [29]. This early osmotic edema is pathophysiologically distinct from vasogenic edema, lacking the prolonged, barrier-disruptive processes associated with tumor growth [29].

These findings are consistent with prior studies suggesting that larger tumor size correlates with more extensive vasogenic edema due to enhanced angiogenic activity and chronic blood–brain barrier disruption [26,27]. The observed dependency of PHE importance on lesion size in our XAI model reinforces its utility in capturing these nuanced pathophysiological differences. This underscores the potential of explainable AI to not only enhance diagnostic accuracy but also provide insights into the underlying biological mechanisms.

Lastly, there is a significant difference in average background importance between the neoplastic and non-neoplastic groups. This finding can be observed in the full study population and in the subgroups of different ICH volumes. The Grad-CAM++ method’s smoothing effect during scaling extends attribution scores slightly beyond the lesion border into adjacent background areas. It seems that the model is paying more attention to the edges of the lesion in neoplastic cases, which leads to higher average importance in the background. This interpretation is consistent with known imaging characteristics of intra-axial brain tumors where “radial or finger-like” extensions or irregular shapes of the PHE can suggest neoplastic etiologies [30]. This visual observation aligns with clinical understanding and provides a plausible explanation for why the background area might show importance in distinguishing between neoplastic and non-neoplastic cases.

This study has some limitations. First, the analysis is performed on a single classifier. Expanding the analysis to multiple classifiers would allow us to assess whether the patterns observed here generalize across different models or if alternative classifiers might focus on different imaging features. For this, new classifiers would have to be developed and tested first. In addition, potential clinical confounders such as tumor histology (e.g., primary vs. metastatic origin) and time from symptom onset to imaging were not included in the analysis. These variables may influence the extent and appearance of perihematomal edema (PHE), potentially affecting model interpretation. However, our model was intentionally designed as a radiology-based decision-support tool that operates on admission non-contrast CT alone, reflecting real-world scenarios in which clinical data may be unavailable or unreliable. In particular, the timing of symptom onset is often unclear in patients with neoplastic disease, especially in older adults. While incorporating such clinical variables could enhance diagnostic specificity, it may also limit model applicability. Future work should explore multimodal approaches that integrate imaging with clinical and temporal data to further refine model performance and interpretability.

Second, our preprocessing approach is masking the image background, effectively forcing the model to focus solely on the regions of interest. From an XAI perspective, it would be interesting to see whether a classifier also considers other regions as relevant to its decision-making. However, our preprocessing was specifically designed to enhance model performance, enabling it to focus on clinically meaningful areas and leverage the capabilities of deep learning-based segmentation. This approach aligns with the goal of maximizing predictive accuracy but prevents an analysis of other regions.

## 5. Conclusions

Our study demonstrates that our previously introduced deep learning model effectively uses both PHE and ICH regions to discern neoplastic from non-neoplastic ICH on admission CT. This suggests that the model’s diagnostic process is grounded in relevant image features rather than incidental associations. Further, the results underscore the ICH region’s predominant influence on the model’s differentiation of ICH types.

## Figures and Tables

**Figure 1 cancers-17-02502-f001:**
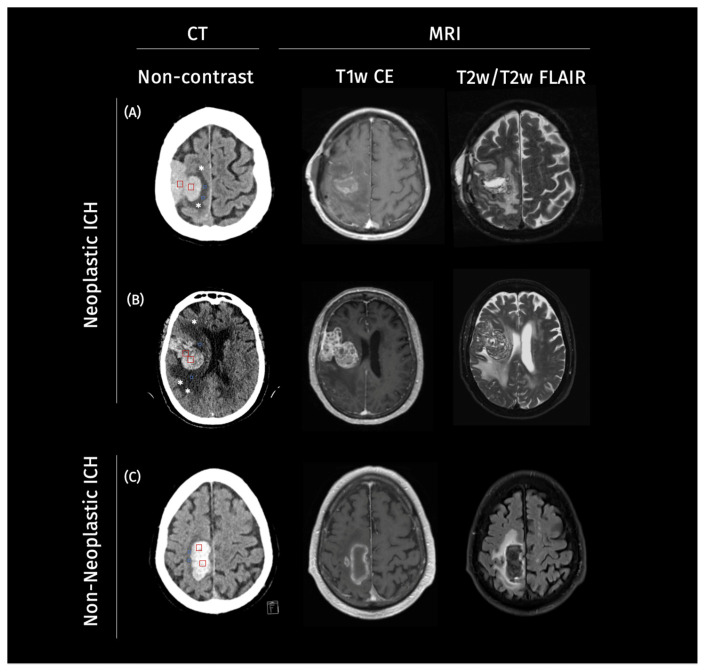
Example cases of neoplastic and non-neoplastic intracerebral hemorrhage on imaging. Legend: Representative axial non-contrast CT images from three patients. On the CT scans, square ROIs were manually placed within the ICH (red) and PHE (blue) regions (two per structure per case) to estimate mean CT density values for each compartment: (**A**,**B**) two neoplastic intracerebral hemorrhages (ICH) demonstrating a heterogeneous lesion, with surrounding irregular perihematomal edema (PHE) marked with a star (*), and (**C**) a non-neoplastic ICH showing a homogenous hyperdense hemorrhage with well-defined margins and symmetric PHE. In Case (**A**), ICH appears mildly hyperdense with a mean of 40–50 HU and mean PHE density of 19–25 HU. Case (**B**) in particular shows an irregular, radially extending edema pattern suggestive of infiltrative tumor growth, with a mean PHE density of 20–29 HU and a mean ICH density of 45–55 HU, in contrast to the more compact and well-defined margins of PHE observed in (**C**), with a mean density of 18–24 HU and, in turn, higher mean ICH density of 60–70 HU.

**Figure 2 cancers-17-02502-f002:**
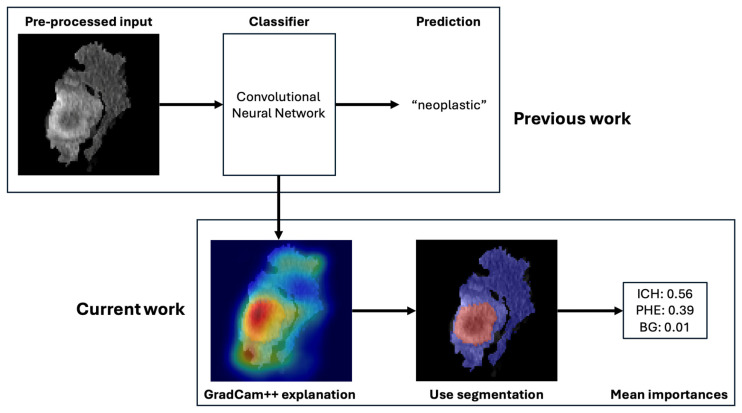
Overview of the deep learning-based classification pipeline. Legend: Workflow of the automated classification system distinguishing neoplastic from non-neoplastic ICH. The process includes (1) input of non-contrast CT scans, (2) segmentation of ICH and PHE regions using a pre-trained nnU-Net model, (3) cropping of regions of interest, and (4) classification via a ResNet-based convolutional neural network. Explainable AI (XAI) methods, including GradCAM++, are applied to interpret predictions.

**Figure 3 cancers-17-02502-f003:**
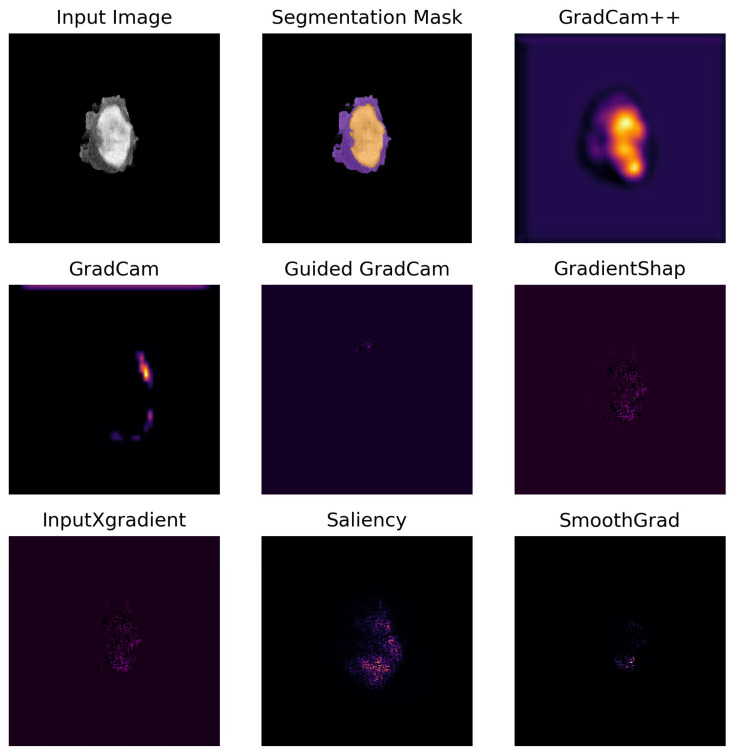
Visualization of each attribution method for one representative case. Legend: Side-by-side visual comparison of attribution maps generated by each explainability method for one representative example case, alongside the input image and the segmentation mask. In the segmentation mask, the perihematomal edema (PHE) is shown in purple, while the intracerebral hemorrhage (ICH) is shown in yellow. In the attribution maps, a brighter yellow color denotes higher importance values, while a darker purple color represents lower importance values.

**Figure 4 cancers-17-02502-f004:**
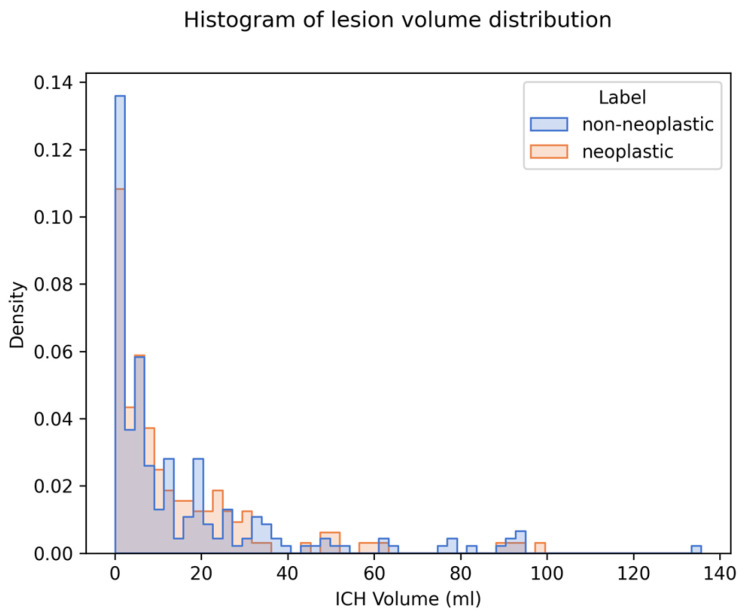
Distribution of ICH volume and density on imaging in the study population. Legend: Histogram displaying the distribution of ICH volumes for both neoplastic (blue) and non-neoplastic (orange) cases. Median ICH volume across both groups was 6.92 mL. There was no statistically significant difference in median volume between the two groups (*p* = 0.477).

**Figure 5 cancers-17-02502-f005:**
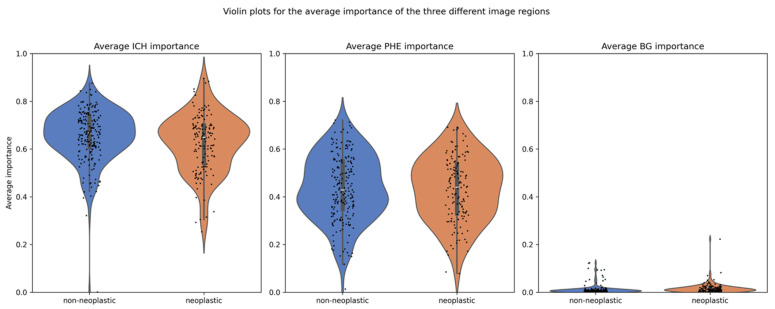
Distribution of importance scores for ICH and PHE regions. Legend: Violin plots showing the distribution of average importance scores (from GradCAM++) for the ICH and PHE regions, stratified by class. The ICH region exhibits significantly higher mean importance than the PHE region (*p* < 0.001), with ICH importance also significantly differing between neoplastic and non-neoplastic cases. PHE importance differs significantly only when stratified by lesion size.

**Table 1 cancers-17-02502-t001:** Baseline characteristics of patients with acute neoplastic and non-neoplastic intracerebral hemorrhage.

	All ICH (n = 349)	Neoplastic ICH (n = 144)	Non-Neoplastic ICH (n = 205)	*p*-Value
Age (years), median (IQR)	67 (53; 78)	66.5 (53; 78)	67 (53;78)	0.998
Female, n (%)	167 (47.85)	69 (47.91)	98 (47.80)	0.983
*Δ* symptom onset to imaging (days), median (IQR)	0.46 (0.13; 1.71)	0.95 (0.2; 5.0)	0.32 (0.09;1.0)	0.013
Hypertension, n (%)	157 (44.99)	39 (27.08)	118 (57.56)	<0.001
CAA, n (%)	49 (14.04)	-	49 (23.90)	-
Oral anticoagulation, n (%)	10 (2.87)	-	10 (4.88)	-
Vascular malformation, n (%)	63 (18.05)	-	63 (30.73)	-
Metastasis, n (%)	102 (29.23)	102 (70.83)	-	-
Tumor, n (%)	42 (12.03)	42 (29.17)	-	-
Median ICH volume, mL (IQR)	6.92 (2.12; 19.91)	7.44 (2.50; 20.26)	6.53 (1.76; 19.56)	0.477
Median PHE volume, mL (IQR)	16.45 (7.01; 39.71)	23.94 (14.24; 64.30)	10.47 (5.32; 24.37)	<0.001

**Table 2 cancers-17-02502-t002:** Faithfulness scores for all tested explanation methods.

	Mean	Standard Deviation
Saliency	0.473	0.047
InputXGradient	−0.256	0.181
SmoothGrad	0.233	0.077
Gradient Shap	−0.092	0.258
GradCam	0.116	0.197
GradCam++	0.49	0.153
Guided GradCam	0.031	0.051

**Table 3 cancers-17-02502-t003:** Average importance for the two classes—neoplastic and non-neoplastic—and the three regions—intracerebral hemorrhage (ICH), perihematomal edema (PHE) and background (BG)—for the full study population and for the subgroups of small and large lesions separately.

Lesion Size	Region	Neoplastic	Non-Neoplastic	*p*-Value
All	ICH	0.615	0.663	<0.001
All	PHE	0.430	0.439	0.54
All	BG	0.017	0.011	<0.001
Small	ICH	0.684	0.717	0.002
Small	PHE	0.471	0.521	0.023
Small	BG	0.013	0.008	<0.001
Large	ICH	0.561	0.600	0.008
Large	PHE	0.396	0.341	0.001
Large	BG	0.019	0.014	<0.001

Legend: The table shows the average importance scores per lesion size, region and class. The intracerebral hemorrhage (ICH) region has higher importance on average, and it is significantly different between the two classes. There is no significant difference between the classes in the perihematomal edema (PHE) region for the full cohort, but there is for the small and large lesions separately. The BG region is significantly different between neoplastic and non-neoplastic cases in both subgroups and the full study population. Generally, for Grad-CAM++ or similar gradient-based attribution methods, a smaller region with non-zero pixels will tend to concentrate importance due to reduced competition for relevance across the image. This is due to scaling that is performed within the method. Therefore, the average importance is higher for smaller lesions.

## Data Availability

The datasets that support the findings of our study are available upon reasonable request from the corresponding author; however, prior approval of proposals may apply by our institution’s data security management, and a signed data sharing agreement will then be approved.

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
