# Peer review of "An Explainable AI Exploration of the Machine Learning Classification of Neoplastic Intracerebral Hemorrhage from Non-Contrast CT"

_cancers, 2025, doi:10.3390/cancers17152502_

Round 1

Reviewer 1 Report

Comments and Suggestions for Authors

This paper provides explainable AI in neuroimaging by revealing how a deep learning model differentiates neoplastic from non-neoplastic intracerebral hemorrhages using non-contrast CT scans.. However, to strengthen the review further and provide a more comprehensive scientific contribution, I recommend the following revisions:

  • The authors should address the potential overfitting of the model to these data sources and consider testing performance on more heterogeneous, real-world datasets.
  • Although Grad-CAM++ yielded the highest faithfulness score, the biological interpretability of the attribution results remains insufficiently discussed. Specifically, it is unclear how the attributed regions correspond to known pathophysiological processes beyond general density and edema. Deeper analysis of biological plausibility is warranted.
  • The manuscript does not explore cases where the model failed. An analysis of false positives and false negatives—particularly whether misclassifications occur in clinically critical scenarios—would be valuable for understanding the limitations of the approach.
  • A discussion on how the results might generalize across imaging modalities is essential.
  • Paper has several typographical and grammatical errors.

Author Response

Authors’ Answers to Reviewer 1 Comments

We sincerely thank the reviewer for their thoughtful and constructive feedback. We appreciate the positive assessment of our work and the helpful suggestions to improve its scientific clarity and completeness. Below, we address each point in turn.

This paper provides explainable AI in neuroimaging by revealing how a deep learning model differentiates neoplastic from non-neoplastic intracerebral hemorrhages using non-contrast CT scans. However, to strengthen the review further and provide a more comprehensive scientific contribution, I recommend the following revisions:

Comment 1: The authors should address the potential overfitting of the model to these data sources and consider testing performance on more heterogeneous, real-world datasets.

Answer 1: Before addressing the reviewer’s comment, we would like to clarify that the training and testing of the deep learning model were conducted as part of a previous study by Nawabi et al (https://doi.org/10.1016/j.imu.2025.101633). The purpose of the current study is solely to evaluate an explainable AI (XAI) design. To enhance the reader’s understanding, we have now included relevant information from that earlier work in the revised manuscript.

In response to the reviewer’s comment, we would first like to appreciate the important point. The potential for overfitting and the importance of training on heterogeneous, multi-center data were thoroughly addressed in our previous work, which focused on model development and evaluation. In that study, we included an external validation cohort from a second academic center and discussed the benefits of broader generalization and multi-institutional data sources.

However, we acknowledge that the distinction between the current explainability study and the prior performance-focused publication may not have been sufficiently clear. To address this, we have revised the Abstract, Introduction, and Study Population sections to more explicitly differentiate between the two studies. We also provided a summary of key model training and evaluation details in the Supplementary Material to ensure this context is easily accessible to readers.

Comment 2: Although Grad-CAM++ yielded the highest faithfulness score, the biological interpretability of the attribution results remains insufficiently discussed. Specifically, it is unclear how the attributed regions correspond to known pathophysiological processes beyond general density and edema. Deeper analysis of biological plausibility is warranted.

Answer 2: We thank the reviewer for this important comment. In response, we have expanded the Discussion section to more explicitly link the attribution results to known pathophysiological mechanisms. Our analysis demonstrated that the highlighted regions correspond to well-established biological processes associated with ICH etiology. Specifically, the model’s focus on ICH density differences aligns with prior findings that neoplastic hemorrhages often appear less dense due to tumor tissue admixture and slower bleeding. The observed differences in PHE importance, particularly its higher relevance in larger neoplastic lesions probably reflect the pathophysiology of vasogenic edema driven by tumor-induced angiogenesis and chronic blood-brain barrier disruption. Conversely, smaller neoplastic lesions showed lower PHE importance, consistent with acute osmotic edema mechanisms. We have also discussed the increased attribution to background areas in neoplastic cases, which we interpret as reflecting tumor-associated irregular lesion margins (e.g.”radial” or “finger-like” extensions), a known imaging feature of neoplastic brain lesions.

Together, these observations support the biological plausibility of the model’s behavior and highlight the capacity of explainable AI to capture clinically and pathophysiologically meaningful patterns beyond traditional manual features.

Comment 3: The manuscript does not explore cases where the model failed. An analysis of false positives and false negatives—particularly whether misclassifications occur in clinically critical scenarios—would be valuable for understanding the limitations of the approach.

Answer 3: Thank you for this suggestion. A detailed case-by-case analysis of false positives and false negatives was performed as part of the original model development study, where we examined potential sources of misclassification and their clinical implications. We kindly invite the reviewer to consult our previous study in detail (https://doi.org/10.1016/j.imu.2025.101633), as well as the additional information provided in the Supplementary Material, where we have included a concise summary of the earlier work to support the context of the current analysis.

In the current study, we focus solely on model explainability rather than performance analysis. As mentioned in comment 1, we made changes throughout the manuscript to specify the scope of this work more clearly.

Comment 4: A discussion on how the results might generalize across imaging modalities is essential.

Answer 4: Thank you for this comment. While our model was trained exclusively on non-contrast CT, we note that future work should explore whether similar patterns of attribution emerge with other modalities, such as contrast-enhanced CT or MRI sequences (e.g., FLAIR weighted-, T1 post-contrast weighted images). For the purpose of this study, we focused on admission non-contrast CT because it is the standard first-line imaging modality in the acute evaluation of ICH. In emergency settings, CT is widely available, fast, and highly sensitive for detecting acute hemorrhage, making it the primary diagnostic tool for initial triage.

We would also like to re-emphasize that the goal of the presented model is to serve as a clinical decision-support tool for identifying neoplastic ICH cases that require urgent and targeted follow-up. This includes timely therapeutic intervention and further diagnostic steps, such as MRI-based etiological characterization and whole-body staging. In contrast, non-neoplastic hemorrhages—such as those due to hypertensive causes—may not require further imaging evaluation. Misclassifying a neoplastic hemorrhage as non-neoplastic at this early triage stage could result in a clinically significant delay or misdiagnosis.

To fully evaluate the model’s potential in MRI-based classification, it would likely be necessary to pursue a more nuanced differentiation—such as distinguishing between types of metastases (e.g., breast cancer vs. melanoma) which falls outside the scope of the present work.

Comment 5: Paper has several typographical and grammatical errors.

Answer 5: We apologize for this inconvenience. We have carefully proofread the entire manuscript and hopefully corrected all errors.

Reviewer 2 Report

Comments and Suggestions for Authors
  1. The abstract lacks clarity on the external validation cohort's specifics, such as sample size and geographical representation, which undermines the generalizability claim. Specify validation cohort demographics and imaging protocols.
  2. Methodological description of the ML model training is insufficient. Missing details include hyperparameters, training-validation-test split ratios, and performance metrics during model development. Add a dedicated sub-section on model training protocols.
  3. The faithfulness metric uses a zero baseline for pixel replacement, but this may not reflect clinical reality. Validate results with alternative baselines (e.g., mean pixel value) to assess robustness.
  4. Observer variability in semi-manual segmentation is not quantified. Report inter-rater reliability metrics (e.g., Cohen's kappa) for ICH and PHE volume measurements.
  5. The subgroup analysis by lesion size (median volume) lacks biological justification. Explain why median volume is a clinically relevant threshold for differentiating neoplastic vs. non-neoplastic ICH.
  6. The discussion fails to address potential confounders like tumor histology (primary vs. metastatic) and time from symptom onset to imaging. These variables may influence edema formation and should be analyzed.
  7. Reference [104] is incomplete (missing author names and title). Ensure all references adhere to the journal's citation style consistently.
  8. The statement that "PHE importance did not outperform ICH" contradicts prior studies (cited as [12,14]) without sufficient mechanistic explanation. Elaborate on why the model prioritizes ICH features over PHE.
  9. The background (BG) importance analysis mentions "finger-like extensions" but lacks quantitative imaging correlates. Include visual examples or radiomic features supporting this interpretation.
  10. The limitation regarding single-classifier analysis is valid, but the study should also acknowledge potential bias from using a single institution's data (Charité) for model development. Discuss multi-center validation plans.
  11. The conclusion overstates the clinical utility by claiming "grounded in relevant image features" without direct comparison to radiologist performance. Add a benchmark against expert interpretation.
  12. Figures 1 and 3 lack detailed captions explaining imaging characteristics (e.g., HU density differences). Enhance visual aids with quantitative annotations.

Author Response

Authors’ Answers to Reviewer 2 Comments

We sincerely thank the reviewer for their detailed and thoughtful feedback. Your comments helped us recognize that we did not make sufficiently clear what distinguishes the current study from our previously published work. Specifically, we understand that some of the concerns relate to aspects of the classification model that were already addressed in the prior study. We would like to clarify that the training and testing of the deep learning model were conducted as part of a previous study by Nawabi et al (https://doi.org/10.1016/j.imu.2025.101633). The purpose of the current study is solely to evaluate an explainable AI (XAI) design. To enhance the reader’s understanding, we have clarified the scope and focus of the present work throughout the main manuscript – specifically in the Abstract, Introduction and Methods. In addition, we have added a summary of the most relevant information about the model training and evaluation to the Supplementary Material, so that readers have easy access to key details without needing to refer to the original paper.

Comment 1: The abstract lacks clarity on the external validation cohort's specifics, such as sample size and geographical representation, which undermines the generalizability claim. Specify validation cohort demographics and imaging protocols.

Answer 1: We appreciate this comment. As mentioned above, this study builds on previously published work, the specifics of the patient cohorts (e.g., demographics, imaging protocols) are detailed in that publication. In the current study, we analyze a pooled cohort comprising two cohorts, as described in Table 1. To improve clarity, we have now revised the Study Population section to explicitly state this.

Comment 2: Methodological description of the ML model training is insufficient. Missing details include hyperparameters, training-validation-test split ratios, and performance metrics during model development. Add a dedicated sub-section on model training protocols.

Answer 2: We thank the reviewer for highlighting this point. As the machine learning model was developed in a previous study, full training details—including architecture, hyperparameters, and evaluation metrics—are presented in that publication. To avoid confusion, we revised the Abstract and Introduction to more clearly state that this study focuses on the analysis of a previously trained model. Nonetheless, we fully agree that this information is important for readers of the current work. Therefore, we have added a concise summary of the training protocol to the Supplementary Material.

Comment 3: The faithfulness metric uses a zero baseline for pixel replacement, but this may not reflect clinical reality. Validate results with alternative baselines (e.g., mean pixel value) to assess robustness.

Answer 3: We appreciate the reviewer’s attention to detail and their thoughtful observation as this is a well-considered point that highlights an important aspect of model interpretability. We agree that the choice of baseline is important and an interesting research question of itself. It is true that a pixel value of 0 does not necessarily reflect clinical reality. However, in our specific case it aligns with the model’s input structure: the classifier was trained on cropped, segmented images where the background was explicitly masked with zeros. As a result, zero represents the absence of information in our pipeline, making it a natural and consistent baseline.

Moreover, because the background area makes up the largest portion of the images, the global mean pixel value in these inputs is already close to zero. Therefore, we don’t anticipate a significant change in the faithfulness metric when using the global mean instead of 0.

Using region-specific means, the mean of the non-zero values or a randomly sampled from the value distribution as baselines introduces additional complexity without clearly improving clinical interpretability and may produce counterintuitive effects—particularly in the masked background.

We chose not to experiment with multiple baselines for two reasons: (1) baseline optimization is not the primary objective of this study, and (2) our focus was on bridging model behavior with clinical reasoning, rather than emphasizing technical evaluation, given the medical journal audience.

That said, if the reviewer believes a baseline comparison would materially strengthen the paper, we are happy to include this analysis in a revised version.

Comment 4: Observer variability in semi-manual segmentation is not quantified. Report inter-rater reliability metrics (e.g., Cohen's kappa) for ICH and PHE volume measurements.

Answer 4: Thank you for this comment. The quality of the segmentations was topic of our previous work, where we focused on deep learning-based segmentation. In that study, the Dice Similarity Coefficient (DSC) was used to evaluate segmentation quality, which is more appropriate for this task than Cohen’s kappa, as DSC directly assesses spatial overlap between segmentations rather than categorical agreement.

The DSC of the model is high for the ICH region with a mean of 0.87 ± 0.02 for the deep-learning model and 0.83 ± 0.05 for the human reference. The intra-rater variability was used as human reference. That is, one rater segmented the same set of images twice and those segmentations were evaluated with the DSC. For the PHE region, the mean DSC of the model was 0.67 ±0.02 and 0.67 ± 0.07 for the human reference[1].

Moreover, we showed that the quality of our manual segmentations with regard to inter- as well as intra-reader variability is high[2]. We included the information in the Supplementary Material to be easily found but refer to the original publications for greater detail.

[1] https://academic.oup.com/radadv/article/2/2/umaf012/8090148?login=false

[2] https://pmc.ncbi.nlm.nih.gov/articles/PMC9844445/

Comment 5:  The subgroup analysis by lesion size (median volume) lacks biological justification. Explain why median volume is a clinically relevant threshold for differentiating neoplastic vs. non-neoplastic ICH.

Answer 5: Thank you for this comment. Currently, there is no established clinical threshold for ICH volume identifying a clinically relevant bleeding and this remains an open research question. In fact, we do have a paper addressing this research gap in a peer-reviewed manuscript evaluation that just got accepted and will be available to the community in “International Journal of Cancer”. For the purpose of this study, we chose the median lesion volume as a pragmatic cutoff to create two balanced subgroups and to explore whether volume-dependent patterns of feature importance might exist. Our findings suggest that lesion size may indeed influence the model’s use of perihematomal edema, and we see this as a first step toward identifying potential volume-related effects. Importantly, we believe that volume thresholds derived from spontaneous ICH studies are not directly transferable to this mixed-etiology cohort.

Comment 6: The discussion fails to address potential confounders like tumor histology (primary vs. metastatic) and time from symptom onset to imaging. These variables may influence edema formation and should be analyzed.

Answer 6: We appreciate the reviewer’s important observation regarding potential confounders such as tumor histology (primary vs. metastatic) and time from symptom onset to imaging.

We would like to clarify that the goal of the current study was not to predict the precise underlying etiology of the hemorrhage, but rather to develop a radiology-based decision support tool that can guide appropriate and timely follow-up imaging. Our model is based solely on non-contrast CT images and does not incorporate clinical or temporal variables. This design choice was intentional, as it reflects real-world scenarios where initial diagnostic decisions often need to be made rapidly and based only on imaging. Regarding the time from symptom onset to imaging, we acknowledge that this variable may influence edema development. However, this information is often unavailable or highly unreliable—particularly in patients with neoplastic disease, who may present with vague or delayed symptoms, and especially among elderly individuals. As a result, incorporating time-to-imaging data would introduce additional uncertainty and limit the generalizability of the model.

We have now explicitly acknowledged this limitation in the revised manuscript. Moreover, we agree that a future multimodal model that integrates imaging with clinical and temporal information may further enhance predictive accuracy. Such approaches are currently being explored in follow-up research.

Finally, we would like to emphasize that the primary aim of this study was to evaluate the explainability and clinical plausibility of the model's predictions (XAI), as this is critical for building user trust and acceptance in clinical practice.

We added the following paragraph to the limitations in the discussion: In addition, potential clinical confounders such as tumor histology (e.g., primary vs. metastatic origin) and time from symptom onset to imaging were not included in the analysis. These variables may influence the extent and appearance of perihematomal edema (PHE), potentially affecting model interpretation. However, our model was intentionally designed as a radiology-based decision-support tool that operates on admission non-contrast CT alone, reflecting real-world scenarios in which clinical data may be unavailable or unreliable. In particular, the timing of symptom onset is often unclear in patients with neoplastic disease, especially in older adults. While incorporating such clinical variables could enhance diagnostic specificity, it may also limit model applicability. Future work should explore multimodal approaches that integrate imaging with clinical and temporal data to further refine model performance and interpretability.

Comment 7: Reference [104] is incomplete (missing author names and title). Ensure all references adhere to the journal's citation style consistently.

Answer 7: We thank the reviewer for pointing this out. The current manuscript includes a total of 30 references, so we believe there may have been a misnumbering in the comment regarding “reference [104].” Nevertheless, we have carefully re-checked all references for completeness and ensured that they fully adhere to the journal’s citation style.

Comment 8: The statement that "PHE importance did not outperform ICH" contradicts prior studies (cited as [12,14]) without sufficient mechanistic explanation. Elaborate on why the model prioritizes ICH features over PHE.

Answer 8: Thank you for this important observation. We agree that our findings initially appear to contrast with prior studies, including our own, that highlighted the diagnostic value of PHE volume in distinguishing neoplastic from non-neoplastic ICH. We have expanded and restructured the Discussion to clarify this. One possible explanation is that neoplastic hemorrhages often present with lower density due to intermixed tumor tissue and slower or pre-existing bleeding associated with ruptures of fragile tumor neoangiogenesis, in contrast to the abrupt vessel rupture typical of non-neoplastic ICH as described in the avalanche model. As a result, the model may place higher importance on CT attenuation patterns within the hemorrhage itself, making ICH density and texture key discriminative features.

It is also important to note that our previous studies used manually extracted imaging features, including absolute and relative PHE volume and mean ICH density. In those analyses, PHE features were indeed highly informative, but their predictive value increased significantly when ICH density was also included in the model. This suggests that PHE alone may not be sufficient for accurate classification and that its diagnostic value may be context-dependent. These segmentations, among additional cases, were used as input for the follow-up machine learning model utilizing a radiomics approach. Radiomics models rely on explicitly defined features, such as PHE volume, shape, or texture, that are manually extracted and often weighted based on known clinical relevance. As such, PHE frequently emerges as a key variable in these models, particularly in the context of neoplastic hemorrhages where vasogenic edema is prominent.

In contrast, deep learning models operate on raw image data and learn hierarchical features automatically. These models may focus more on complex local texture, intensity gradients, or irregular densities, overall representing features more commonly concentrated within the ICH itself, especially in cases of tumor infiltration. Therefore, while PHE may remain clinically important, the model may find stronger or more discriminative signals within the hematoma region. This distinction underscores the complementary strengths of radiomics and deep learning: the former offers transparency through predefined features, while the latter can capture subtle image patterns that may elude traditional quantification.

Comment 9: The background (BG) importance analysis mentions "finger-like extensions" but lacks quantitative imaging correlates. Include visual examples or radiomic features supporting this interpretation.

Answer 9: We thank the reviewer for this helpful comment. The term “finger-like extensions” is commonly used in oncologic imaging to describe irregular, radially spreading edema patterns—particularly in cases where the tumor is located adjacent to the cortex and the edema extends along the U-fibers[1, 2]. While this term serves as a visual and descriptive analogy, we agree that it should be used cautiously and more descriptively in the manuscript.

Accordingly, we have revised the manuscript to describe the edema morphology in more general terms (e.g., “irregular or radially extending edema patterns”) and have removed or softened the phrase “finger-like” to avoid overinterpretation. In addition, we have updated Figure 1 to better illustrate this visual pattern. Specifically, Case 1B shows this morphology clearly, in contrast to Case 1C, where such features are absent. We believe these visual examples help clarify the observed differences in background importance and support the interpretation within the framework of known imaging patterns in neoplastic brain lesions.

[1] https://doi.org/10.1186/s12957-015-0496-7

[2] https://doi.org/10.3892/ol.2015.3639

Comment 10: The limitation regarding single-classifier analysis is valid, but the study should also acknowledge potential bias from using a single institution's data (Charité) for model development. Discuss multi-center validation plans.

Answer 10: We completely agree that bias can be introduced when using data from a single institution. In fact, we discussed this limitation in the previous work in which we described the model’s training and performance in detail. Please find a short summary of this in the Supplementary Material and refer to a more detailed discussion about the limitations and future research in the original publication.

Comment 11: The conclusion overstates the clinical utility by claiming "grounded in relevant image features" without direct comparison to radiologist performance. Add a benchmark against expert interpretation.

Answer 11: Thank you for this observation. We agree that direct comparison to expert interpretation is essential for assessing clinical utility. As noted, this study focuses on explainability rather than performance benchmarking. However, the underlying model was previously evaluated against an expert radiologist, and its performance was found to be comparable. We added a remark about the performance evaluation of the model in Supplementary Material. A comparison between the importance of imaging features for humans versus AI would also be a very interesting future research question. As it is difficult for humans to state precisely which parts of the images they used for their decision, eye tracking might be employed to analyze the gaze of the radiologists while they make their diagnosis.

Comment 12: Figures 1 and 3 lack detailed captions explaining imaging characteristics (e.g., HU density differences). Enhance visual aids with quantitative annotations.

Answer 12: Thank you for this helpful suggestion. Informative figures are key to a good publication. We have revised the captions for Figures 1 and 3 to include more detailed descriptions of the imaging characteristics. Please note that Figure 3 is now Figure 4 as we added a new figure to the manuscript.

We extensively revised Figure 1 adding descriptions to the density characteristics as well as information on the PHE shape:

Figure 1. Example Cases of Neoplastic and Non-Neoplastic Intracerebral Hemorrhage on Imaging.

Legend: Representative axial non-contrast CT images from three patients. On the CT scans, square ROIs were manually placed within the ICH (red) and PHE (blue) regions (two per structure per case) to estimate mean CT density values for each compartment: (A and B) Two neoplastic intracerebral hemorrhages (ICH) demonstrating a heterogeneous lesion with surrounding irregular perihematomal edema (PHE) marked with a star (*), and (C) a non-neoplastic ICH showing a homogenous hyperdense hemorrhage with well-defined margins and symmetric PHE. In Case (A) ICH appears mildly hyperdense with a mean of 40–50 HU and mean PHE density of 19–25 HU. Case (B) in particular shows an irregular, radially extending edema pattern suggestive of infiltrative tumor growth with also mean PHE density of 20–29 HU and a mean ICH density of 45–55 HU, in contrast to the more compact and well-defined margins of PHE edema observed in (C) with a mean density of 18-24 HU and in turn higher mean ICH density of 60–70 HU.

Reviewer 3 Report

Comments and Suggestions for Authors

Recommendation

Comments:

The manuscript describes using explanable artificial intelligence (xAI) methods for enhancing transparency and interpretability of deep learning models for medical imaging.

I recommend the article for publication upon addressing the comments listed here:

  1. Regarding the importance of the article, this article performed GradCam visualizations on a very small dataset of cancer images, and using only one type of classifier. I am not convinced of the novel contribution of the article as the main conclusion is that “the model's diagnostic process is grounded in relevant image features rather than incidental associations. Further, the results underscore the ICH region’s predominant influence on the model’s differentiation of ICH types”. In my experience with computer vision, these are evident due to the different contrast patterns of ICH vs. PHE regions. And there has been many studies using CNN for segmentation of Intracerebral hemorrhage.
  2. In section 2.4, the authors mentioned that Explanations are generated using various gradient visualization methods and refers to a comprehensive description of these methods in the Supplementary Information. First of all, I am unable to find the Supplementary Information, so perhaps the author need to reupload the Supplementary Information with the above information. Secondly, I would suggest moving the comprehensive information to the Introduction part because they are relevant as literature background information for the article.
  3. In addition to comment 1, which is more relevant to literature background. Another comment is for section 2.4, in terms of results, we don’t get to see in the main manuscript anywhere the comparison of GradCam vs. GradCam++ vs. Others. I would recommend to plot the feature importance visualizations side by side for the different methods that the authors have tested in the results section. Maybe the authors plotted these in the Supplementary Information (which I have no access to), I suggest to move them in the main manuscript.
  4. Again in section 2.5, the results Table 2 in Supplementary Information need to be in the main manuscript, at this point, I am not at all convinced that GradCam++ is the best method because I have not seen the result plots in the Supplementary Information.

Author Response

Authors’ Answers to Reviewer 3 Comments

First and foremost, we sincerely thank the reviewer for taking the time to review our manuscript and for providing thoughtful and constructive comments.

We would like to clarify that the training and testing of the deep learning model were conducted as part of a previous study by Nawabi et al (https://doi.org/10.1016/j.imu.2025.101633). The purpose of the current study is solely to evaluate an explainable AI (XAI) design. To enhance the reader’s understanding, we have clarified the scope and focus of the present work throughout the main manuscript – specifically in the Abstract, Introduction and Methods. In addition, we have added a summary of the most relevant information about the model training and evaluation to the Supplementary Material, so that readers have easy access to key details without needing to refer back to the original paper.

We appreciate your feedback and suggestions and apologize for the inconvenience caused by the unavailable supplementary material. Below, we respond point by point to each of your comments.

To prevent any further access issues and avoid potential misunderstandings, we have now incorporated the supplementary material directly into the main manuscript.

Reviewer 3

The manuscript describes using explainable artificial intelligence (xAI) methods for enhancing transparency and interpretability of deep learning models for medical imaging.

I recommend the article for publication upon addressing the comments listed here:

Comment 1: Regarding the importance of the article, this article performed GradCam visualizations on a very small dataset of cancer images, and using only one type of classifier. I am not convinced of the novel contribution of the article as the main conclusion is that “the model's diagnostic process is grounded in relevant image features rather than incidental associations. Further, the results underscore the ICH region’s predominant influence on the model’s differentiation of ICH types”. In my experience with computer vision, these are evident due to the different contrast patterns of ICH vs. PHE regions. And there has been many studies using CNN for segmentation of Intracerebral hemorrhage.

Answer 1: We appreciate the reviewer’s concern and acknowledge that convolutional neural networks have been widely used in ICH segmentation. However, our study is not focused on segmentation or classification model development, but rather on providing a detailed and quantitative explanation of a previously validated classification model's decision-making process using explainable AI (XAI) techniques. While it may appear intuitive that ICH features dominate due to different contrast patterns, our results empirically demonstrate this across 349 clinically diverse cases using region-specific attribution scores and statistical comparisons.

We also emphasize that the importance of PHE varied with lesion size, and that differences between neoplastic and non-neoplastic PHE regions were not uniformly distributed—highlighting subtle and non-obvious patterns and linking them to expert clinical understanding. We have clarified this distinction further in the revised Introduction and Discussion sections.

We also recognize the limitation of using only one classifier in our analysis. We outline this in the Discussion alongside other limitations. We agree that exploring how different model architectures—for example transformer-based models—attribute importance to imaging features would be very interesting. We anticipate that different models might show some variation in attribution patterns, but that similarities are likely because clinically meaningful differentiators between neoplastic and non-neoplastic ICH on non-contrast CT are limited. Unless of course, different models were to highlight features that are yet unknown or invisible to the human eye. This could open exciting new avenues for research.

Comment 2: In section 2.4, the authors mentioned that Explanations are generated using various gradient visualization methods and refers to a comprehensive description of these methods in the Supplementary Information. First of all, I am unable to find the Supplementary Information, so perhaps the author need to reupload the Supplementary Information with the above information. Secondly, I would suggest moving the comprehensive information to the Introduction part because they are relevant as literature background information for the article.

Answer 2: We sincerely apologize for the missing Supplementary Information and any inconvenience this caused. We are unsure what went wrong during submission, but to ensure that the content is fully accessible, we have now appended the supplementary material directly to the end of the manuscript.

As for the detailed descriptions of each explanation method, we agree that background context is important. However, due to limited space and reduced readability, we have chosen to keep the full methodological comparisons in the supplementary section. To address your suggestion, we have now extended our summary of the explanation methods and their key differences in the Methods along with citations, to better situate the reader.

Comment 3: In addition to comment 1, which is more relevant to literature background. Another comment is for section 2.4, in terms of results, we don’t get to see in the main manuscript anywhere the comparison of GradCam vs. GradCam++ vs. Others. I would recommend to plot the feature importance visualizations side by side for the different methods that the authors have tested in the results section. Maybe the authors plotted these in the Supplementary Information (which I have no access to), I suggest to move them in the main manuscript.

Answer 3: Thank you for this helpful suggestion. In response, we have now added a new figure (Figure 3) to the main manuscript that presents a side-by-side visual comparison of attribution maps generated by GradCam, GradCam++, and other methods for one representative case. This allows readers to visually assess the differences in attribution styles and focus areas across methods.

Comment 4: Again, in section 2.5, the results Table 2 in Supplementary Information need to be in the main manuscript, at this point, I am not at all convinced that GradCam++ is the best method because I have not seen the result plots in the Supplementary Information.

Answer 4: We fully agree and thank the reviewer for pointing this out. We have now moved the faithfulness metric results into the main manuscript as Table 2. This table shows that GradCam++ achieved the highest average faithfulness score among all evaluated methods, justifying its selection for subsequent region-wise attribution analysis.

Round 2

Reviewer 1 Report

Comments and Suggestions for Authors

In light of the authors’ detailed responses to the comments, the overall quality of the manuscript has been substantially enhanced. Therefore, the paper is deemed suitable for acceptance and publication in the journal. 

Reviewer 2 Report

Comments and Suggestions for Authors

I agree to accept in this version.